# Digital imaging and vision analysis in science project improves the self-efficacy and skill of undergraduate students in computational work

**Tessa Durham Brooks**[1]*, **Raychelle Burks**[2], **Erin Doyle**[1], **Mark Meysenburg**[3], **Tim Frey**[4]

1 Department of Biology, Doane University, Crete, Nebraska, United States of America, 2 Department of Chemistry, American University, Washington, DC, United States of America, 3 Department of Computer Science, Doane University, Crete, Nebraska, United States of America, 4 Department of Education, Doane University, Crete, Nebraska, United States of America

* tessa.durhambrooks@doane.edu

**Data Availability Statement:** All relevant data are within the manuscript and its Supporting information files.

## Abstract

In many areas of science, the ability to use computers to process, analyze, and visualize large data sets has become essential. The mismatch between the ability to generate large data sets and the computing skill to analyze them is arguably the most striking within the life sciences. The Digital Image and Vision Applications in Science (DIVAS) project describes a scaffolded series of interventions implemented over the span of a year to build the coding and computing skill of undergraduate students majoring primarily in the natural sciences. The program is designed as a community of practice, providing support within a network of learners. The program focus, images as data, provides a compelling 'hook' for participating scholars. Scholars begin the program with a one-credit spring semester seminar where they are exposed to image analysis. The program continues in the summer with a one-week, intensive Python and image processing workshop. From there, scholars tackle image analysis problems using a pair programming approach and can finish the summer with independent research. Finally, scholars participate in a follow-up seminar the subsequent spring and help onramp the next cohort of incoming scholars. We observed promising growth in participant self-efficacy in computing that was maintained throughout the project as well as significant growth in key computational skills. DIVAS program funding was able to support seventeen DIVAS over three years, with 76% of DIVAS scholars identifying as women and 14% of scholars identifying as members of an underrepresented minority group. Most scholars (82%) entered the program as first year students, with 94% of DIVAS scholars retained for the duration of the program and 100% of scholars remaining a STEM major one year after completing the program. The outcomes of the DIVAS project support the efficacy of building computational skill through repeated exposure of scholars to relevant applications over an extended period within a community of practice.

**Funding:** TDB, RB, MM - 1608754 - National Science Foundation - https://www.nsf.gov/awardsearch/showAward?AWD_ID=1608754&HistoricalAwards=false TDB, ED - 1557417 - National Science Foundation - https://www.nsf.gov/awardsearch/showAward?AWD_ID=1557417&HistoricalAwards=false RB - BH-0018 - The Welch Foundation - https://www.welch1.org/ The funders had no role in study design, data collection and analysis, decision to publish, or preparation of the manuscript.

**Competing interests:** The authors have declared that no competing interests exist.

## Introduction

Science, technology, engineering, and mathematics (STEM) professions, even those not traditionally steeped in quantitative models and data analysis, increasingly require computational competence [1]. In particular, the natural sciences have experienced significant increases in the amount of data generated by increased computing power, cheaper and more rapid sequencing technologies, and the rise of interdisciplinary fields such as personalized medicine, phenomics, digital agriculture, and climate science. Computation has become so ubiquitous and necessary across the natural and physical sciences that it has been referred to as the "third pillar of the scientific method," along with theory and experimentation [2]. A career in the natural sciences increasingly requires that professionals are comfortable with basic computational skills and quantitative analysis [3–5]. Beyond this, modern scientific exploration may require the design of new software by developers with both specific content knowledge and computational skills. As a potential "end user", a biologist, chemist, physicist, etc. has the content knowledge, but may need computational skills training [6, 7]. Across the broad range of STEM disciplines, too few students are being trained in computational and quantitative skills that would enable them to develop useful software. In particular, undergraduate students in the life sciences may be resistant to developing quantitative or computational skills due to previous negative experiences or a perception that they "aren't good at" mathematics or computers [8]. The result of these factors is a mismatch between the skills needed for success in research or industry positions and the skills possessed by graduates and young professionals starting these positions.

To address this mismatch, we conceived of the Digital Imaging and Vision Applications in Science (DIVAS) Project. This year-long program was designed as a guided 'onramp' to develop computational skills within a community of practice that would contribute to participants' STEM career success. The overall goal of the DIVAS project is to develop, utilize, and test interventions that will engage and train STEM undergraduate students in computing—especially students that do not traditionally participate in computer science curriculum. DIVAS interventions present students with visually-appealing image-based problems relevant to the disciplines they are majoring in, thereby making the skills we aim to develop eminently practical. Importantly, it is relatively easy to capture images with high spatial, temporal, and spectral resolution, with images being increasingly used as data in scientific, clinical, and engineering settings [9–12]. While images are relatively easy to obtain, extracting useful information from them commonly presents technical barriers that lead to processing bottlenecks. Although the collection of large datasets has become rather commonplace, scientists of various career stages may lack the computational skills to analyze these data independently or may have limited access to productive collaborations with computer scientists or other specialists. Early introduction to computational approaches, along with frequent practice, enables a person new to computing to take advantage of training resources to develop critical skills and to form effective collaborations [13–15]. Studies of computer science courses that present instructional concepts in the context of digital images, videos, or music—i.e. "media computation" [16]—contributes to higher retention of women and non-computer science majors in these courses [13, 17, 18].

Just as computation-in-context supports student gains, so do communities of practice and learning communities. Both types of communities, which can be quite distinct depending on their specific model [19], are often used interchangeably to describe a community for sharing, developing, and/or maintaining knowledge, skills, and practices within which membership ranges from novices to seasoned experts. For students, participation in such communities has been shown to boost academic performance, self-efficacy, sense of belonging, STEM identity,

retention, and graduation rates [20–23]. In the DIVAS Project, cohorts of novices work side-by-side with faculty mentors, and their more experienced student peers, to themselves become more advanced practitioners via legitimate peripheral participation [24]. Importantly, the DIVAS Project models the reality of the modern computational work environment, which is soundly a team-based endeavor. This counters the stereotype that such work is largely solitary.

The general hypothesis of the DIVAS Project is that gradual, scaffolded exposure to—and practice with—computational tools, centered on accessible and relevant applications, and implemented in both simulated and authentic supportive professional environments, will impact student self-efficacy, computational competency, and career path interest and knowledge. We have taken the approach of emphasizing growth in self-efficacy toward computing as the first necessary indicator of growth in computational skill [25–27]. We also posit that as participants become more familiar with computational tools, they will additionally show more interest in career paths that would utilize said tools. Though our pilot program was restricted in size, its positive impact on participants suggests that DIVAS program elements are well-suited to our broader goals of fostering computation skills within a community of practice. We describe our approach here both as a guide and an invitation. We hope to form new DIVAS partnerships to broaden the DIVAS community and enable additional study on the efficacy of the approach we have taken.

## DIVAS program elements

To explore our hypothesis, a pathway of interventions was designed that comprise our programmatic 'onramp' (Fig 1). Each cohort of DIVAS scholars was introduced to our community of practice via a one-credit, spring semester seminar (DIVAS Seminar I) and engagement with the DIVAS Slack team. Work continued in the summer with a week-long coding workshop, followed by a four-week long paired-programming session that allows DIVAS scholars to put their recently acquired skills to use. DIVAS scholars can participate in an additional three weeks of research with DIVAS faculty to conclude their summer activities. In the fall semester, DIVAS scholars returning to research—or starting new computing projects—continue engaging with community members using our Slack team. During the following spring, the cohort takes DIVAS Seminar II. As with other team science endeavors, the DIVAS community is offline and online, with Slack and Zoom playing significant roles in communication, project management, co-working sessions, team meetings, etc. In the sections that follow, the basic design of each intervention is detailed below and intervention resources can be found in S1–S17 Files.

**DIVAS Seminar I and II.** DIVAS Seminar I and II are both one-credit seminars offered in the spring semester. DIVAS Seminar I is offered to new scholars before the summer coding workshop and projects. DIVAS Seminar II is offered to scholars in the spring after they have completed the summer interventions. DIVAS Seminar I is designed to introduce students to

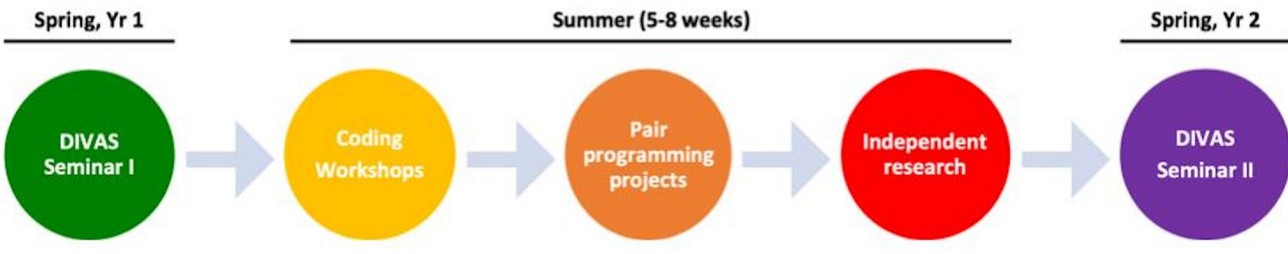

**Fig 1. Interventions comprising the computational 'onramp' of the DIVAS program.**

images as data and basic coding concepts, as well as allow them to meet professionals who use coding in their everyday work and explore computing careers. Students complete a photo journal project where they identify a question or problem of interest, collect a series of images to address that question or problem, then use ImageJ to conduct simple image processing. In the first iteration of DIVAS Seminar II, scholars organized their project code repositories, developed online portfolios regarding their DIVAS experiences, and gave a local conference presentation. Based on student feedback, the next year's Seminar II was modified to include more challenging academic content. Students learned the basics of parallel programming using Python and OpenMPI, creating a "Burning Ship" fractal image [28] using the DU supercomputer, Onyx. The third year, DIVAS Seminar II walked a line between the first two iterations; students worked on cleaning the previous summer's code and keeping the code repository up to date in addition to helping test the new version of the image processing workshop that used the scikit-image processing libraries instead of OpenCV. DIVAS Seminar II was taught by the same instructor each year and scheduled at the same time as DIVAS Seminar I. This scheduling arrangement made it easier to promote interactions between the cohorts in each class. Further developing peer mentorship opportunities, the DIVAS faculty created a "writing center for computing" on campus called the Center for Computing in the Liberal Arts (CCLA) in year two of the project [29]. The center was led by a staff person hired, in part, to serve this role. Upon creation of the CCLA, several DIVAS scholars signed up to serve as peer mentors, assisting in the creation of training materials and participating in center activities.

**Coding workshop.** Short courses, such as those run by The Carpentries, have become a popular way to build coding and data analysis skills [30]. On average, participants report increased self-efficacy in coding and coding skills, based on pre- and post-workshop surveys and on longitudinal surveys [30, 31]. However, workshops like those offered through The Carpentries are not targeted towards, nor significantly attended by, undergraduate students [30]. We designed a one-week coding workshop that includes two days of basic coding in Python and three days of image processing using OpenCV libraries. The two-day introduction to Python was modeled on an existing Carpentries workshop and can be found at GitHub [32]. The overall design of the three-day image processing workshop was informed by Adrian Rosebrook's 2016 book on the topic [33]. To keep students engaged with Python basics, examples used during this section of the workshop were tailored toward image processing projects. Students were also presented with two authentic and "solvable" research problems at the beginning of the image processing portion of the workshop. For the first problem, participants were asked to count bacterial colonies on a plate image. For the second, participants were asked to track the progress of an acid-base titration captured on video. Our workshop design provides students an opportunity to immediately apply their recently acquired Python skills to write code that performs analysis tasks to address these two authentic problems. The image processing portion of the workshop was adopted by The Carpentries in 2019 [31, 34]. At the same time, the image processing operations were translated into Scikit-image, which is much easier to install and implement across a wide range of hardware, software, and network environments. Workshop materials are available at its Data Carpentry site [31].

**Pair programming projects.** Pair programming is a practice used in the software development industry in which two programmers work together, with one person assuming the role of the "driver" who writes the code, and the other taking the role of "observer" who reviews the code and makes suggestions. In introductory computer science courses, the use of pair programming results in higher quality code, increased student enjoyment, improved pass rates for courses, and improved retention in computer science majors for both men and women [17, 35–37]. Also, pair programming has been shown to increase the confidence of women in the programming solutions they produce [35]. We designed the DIVAS program so

that participants would transition from the coding workshop to pair programming work, applying knowledge gained in the workshop to the completion of two consecutive two-week pair programming projects. Each year, one project was morphometric in nature while the other was colorimetric. Image data sets were found from public repositories or from the research of the faculty team. The project was presented by a faculty member at the beginning of each project. Pairs were determined by the faculty facilitators. One pair was composed of the SEU scholar and a DU scholar. For this pair, pair programming was conducted virtually using Zoom. A significant amount of project management was done via the DIVAS Slack team. To promote a community of practice, scholars participated in daily "stand-up" meetings where they gave brief progress reports and set goals for the day. Issues were also shared and discussed. Pairs worked on code for the remainder of the day. A formal code review was conducted each week by the DIVAS community of practice, with community members joining both in-person and virtually. All participants were to have copied and annotated the code of the other teams prior to the review to prepare for discussion. At the conclusion of the review meeting, the group identified major goals for the following week or final items to wrap up the project. The process was repeated with a second project with different partners in the following two weeks. An example of a pair programming project is provided in S8 File.

**Independent research.** In year one, scholars were required to conduct 3–4 weeks of independent research after completing pair programming. Projects were based on the existing research of the faculty team as well as being informed by student interest (Table 1). Students generally worked independently, but met with their faculty advisor for daily check-ins and to troubleshoot any problems that arose. Participating in DIVAS research was optional in years two and three to better accommodate student schedules, e.g. participation in a Research Experience for Undergraduates (REU) program, study abroad, etc.

Within the DIVAS Project framework, several questions were explored: 1) How do program interventions impact participant self-efficacy toward computation? 2) How do program interventions impact participant career interest and knowledge? 3) How do program interventions and their ability to demonstrate effective computational thinking?

The overall objectives of the DIVAS project are to:

1. Explore the effectiveness of coding workshops on student attitudes toward computation and their ability to demonstrate effective computational thinking.

2. Measure impacts of paired programming projects, independent research, and professional development seminars on self-efficacy and ability to apply computational skills.

3. Investigate the impact of interventions in computation on student career interest and knowledge.

**Table 1. Example pair programming and independent research projects.**

| |
|---|
| 1. Detection of breaks in veterinary x-ray images |
| 2. Detection and quantification of standards printed onto a solid surface |
| 3. Calculating the endpoint of a titration from a movie of the reaction |
| 4. Counting plaques on an agar plate |
| 5. Quantifying chemotaxis of bacteria toward potential attractants |
| 6. Measuring growth of maize seedlings over time |
| 7. Automatically analyzing and solving images of printed Sudoku problems |
| 8. Improving script performance by converting code from python to C to improve script performance |

## Materials and methods

### Study context and overview

The DIVAS program was administered at Doane University (DU) and St. Edward's University (SEU). Both DU and SEU are private liberal arts institutions. DU is a rural residential under-graduate campus in Crete, Nebraska. SEU is an urban campus in Austin, Texas with designations as both a minority serving institution and Hispanic serving institution. At the time of the study, DU had approximately 1,070 undergraduates with 45% identifying as women, 17% identifying as a member of an underrepresented minority (URM; African Americans, American Indians including Native Alaskans, Hispanics and Native Pacific Islanders) group, 34% identifying as first-generation university students, and 30% of students being Pell-eligible. The undergraduate population at SEU was approximately 3,445, with 62% identifying as women, 55% identifying as a member of an URM group, 33% identifying as first-generation university students, and 38% of students being Pell-eligible.

For each year of the DIVAS program, up to six scholars were selected. Participant number was limited due to budget constraints. Every effort was made to include each student who completed an application in the DIVAS program. A total of 17 scholars were selected to complete all program interventions and a participant overview is provided in Table 2. One scholar majoring in a chemistry-related field was recruited from SEU each year. One student from DU majoring in computer science (CS) was recruited each year to further build the community of practice by creating additional types of peer interactions. Faculty, students, and staff at the hosting institution were invited to participate in the annual coding workshop. An additional 15 students (47% of whom identified as women) participated in this way and completed pre- and post-surveys.

Baseline scores in self-efficacy of DIVAS scholars (non-CS majors) were compared to two other groups of students. CS majors were excluded due to their higher baseline self-efficacy and because they were not represented in our comparison groups. The first comparison group (comparison group 1) was taken from students enrolled in an introductory chemistry course at DU (CHM 125). CHM 125 is one of the courses DIVAS scholars were recruited from. One section of the course was surveyed each fall. A total of 67 students in CHM 125 completed these surveys, 49% of whom identified as women. The second comparison group (comparison group 2) consisted of 15 additional non-scholar students who participated in the coding work-shop and completed pre- and post-assessments, as mentioned in the previous paragraph. These students had elected to conduct research over the summer and participated in the work-shop to gain computational skills beneficial to their projects.

Each of the three years of the study, the DIVAS program was advertised using flyers (digital and paper), online and social media posts, and visits by faculty and existing scholars to classes that are generally enrolled by first year and sophomore natural science majors. Written, informed consent was obtained from all participants and the study was approved by the Doane University Institutional Review Board (IRB). Self-efficacy and career path data were collected by completing and submitting an electronic survey administered using Qualtrics

**Table 2. Participant overview.**

|  | Year 1 | Year 2 | Year 3 | Total |
|---|---|---|---|---|
| DIVA Scholars | 6 (66%) | 6 (83%) | 5 (80%) | 17 (76%) |
| Coding Workshops | 14 (50%) | 10 (71%) | 10 (58%) | 34 (59%) |

Participants who completed assessments, with % women in parenthesis.

software (Qualtrics, Provo, UT). Pre- and post-assessments for each intervention were completed by participants. The post-assessment data from the prior intervention was used as a baseline for the next intervention in the pipeline. Scholars completed the self-efficacy and career path surveys five times, first as a pre-survey and then after DIVAS Seminar I, the coding workshop, pair programming or summer research, and DIVAS Seminar II. Participants also provided written consent to complete computational thinking prompts and to submit code generated for analysis, which was scored by project researchers. Scripts and/or prompt responses were collected from scholars at four points: before and after DIVAS Seminar I, after the coding workshop, and after pair programming. If scholars completed summer research, the scripts they generated were collected. One scholar left the program after completing DIVAS Seminar I, which reduced the sample size to sixteen for that intervention. Two scholars did not complete surveys at the end of pair programming and summer research and one scholar from year 2 did not complete the survey at the end of DIVAS Seminar II, reducing the sample sizes for these interventions to fourteen and ten, respectively.

## Self-efficacy and career path assessment

A Qualtrics survey was used to measure perceived self-efficacy in computing and knowledge of and interest in career paths involving computing. This survey, titled 'DIVAS Career Path and Self-Efficacy', is based on three previously-designed and validated surveys [38–40]. Survey questions ask participants to score their computational thinking and their ability to use computational tools to solve problems (self-efficacy); to indicate their interest in incorporating computational thinking and CS tools into their careers (career interest); and to indicate how much they know about careers using computer science applications, programming, or computational thinking and where to find this information (career knowledge). Twelve questions related to self-efficacy are answered as a user-inputted number on a 100-point scale, with higher values representing more self-efficacy for a particular item. Seven questions related to career paths (three items related to career interest and four items related to career knowledge) include response choices on a four- or five-point Likert-type scale. The self-efficacy instrument had an internal reliability of 0.95 with no improvement by removal of any single item. The career interest and career knowledge instruments had internal reliabilities of 0.82 and 0.67, also with no improvement in reliability with removal of any single question. Participants took the combined career path (career interest plus career knowledge) and self-efficacy survey before and after the major interventions in the project, as described above. If the participant had previously completed the survey after an intervention, this score was used as the pre-survey for a subsequent intervention.

## Computational thinking assessment

To assess computational thinking ability before any formal instruction in coding, participants were given a handout that described a hypothetical cup stacking robot that could be given simple instructions to achieve different configurations of cups. The exercise was adapted from the Hour of Code lesson "Programming Unplugged: My Robotic Friends" [41]. Participants were asked to create a series of commands to achieve a particular cup stacking arrangement. After writing their initial set of commands, participants were asked to simplify their 'code', possibly by writing one or more new commands. A different cup-stacking prompt was used after the DIVAS Seminar I. After each subsequent intervention, the code developed during each one was used to assess computational thinking ability. The cup stacking prompts are available in S15–S17 Files.

Cup-stacking prompt work and scripts (hereafter 'artifacts') were evaluated using a rubric developed based on definitions from the International Society for Technology in Education (ISTE), Computer Science Teachers Association (CSTA), Carnegie Mellon, Google, and Harvard [42–45]. Our computational thinking rubric was organized into the first four phases of the RADIS (Recognize / Analyze / Design / Implement / Support) framework [46]. The Recognize section measures how well the problem is understood and one's ability to gather the data needed to solve the problem. The Analyze section measures the ability of the participant to understand the options available to solve the given problem. This section also measures the ability of the participant to use abstraction, modeling/representation, and decomposition to design a solution to a problem. The Design section measures the participant's ability to design an effective algorithmic procedure to solve the problem. It includes the participant's ability to use sequence, selection, and iteration. The Implementation section addresses the ability of the participant to transform the algorithm into working code to solve a given problem. It also addresses the evidence that is used, reused, and remixed from previous projects or other sources. Finally, the Implement section assesses any testing or debugging that was used to improve the code. The original rubric was scored on a three-point scale; Proficient (3), Progressing (2), or Novice (1). Subsequent iterations included five levels, first from 0 to 4, then from 1 to 5. The additional levels were added to better accommodate the types of variation we were seeing in the scored artifacts. The expanded scale was adjusted to start with '1' from '0' to make statistical analysis more interpretable. The internal reliability of the first version and final versions of the instrument was high overall (Cronbach's alpha of 0.94–0.95). Interrater reliability (IRR), determined by percent agreement, was 76% using a set of seven artifacts scored by three raters [47]. The reliability of each section for both the first and final versions ranged from a Cronbach's alpha of 0.80 for the 'Implement' section to a Chronbach's alpha of 0.97 for the 'Design' section. The iterations of our rubric and specific changes made in each are available in S10–S14 Files.

## Data analysis

**Self-efficacy and career path assessments.** To investigate changes in student self-efficacy (SE), career path interest (CI) and career knowledge (CK), scores within each category were summed to determine a composite SE, CI and CK score for each individual. A paired-samples t-test was performed (alpha = 0.05) to determine if composite scores before and after a given intervention were significant. For significant changes, the effect size was determined by calculating Cohen's d.

**IDEA survey data.** Voluntary end-of-course evaluations at DU are taken through the IDEA Student Ratings of Instruction system survey [48]. In addition to evaluating the course and instructor, students respond to their perceived learning gains in thirteen pre-defined objectives [49]. Survey responses to these learning gains, along with responses to items related to perceptions of the course difficulty and motivation to take the seminar, were collected as available. Eleven of fourteen (79%) DU scholars completed the IDEA survey after DIVAS Seminar I. Six of seven DU scholars (86%) completed the IDEA survey after DIVAS Seminar II. Median scores for each item were determined along with the percentage of responses that were scored as a '4' or '5' (the two highest rankings) for each item.

**Computational thinking assessments.** Computational thinking (CT) scores were generated by first determining the mode within each criterion for any artifacts scored by two or more people. Select artifacts had more than one scorer to determine and monitor IRR. The median score within each section of the rubric was used to define a subsection score. The median scores were added to give a total CT score. To determine if total CT scores changed,

the following analysis was done: (1) a paired-samples t-test was performed on scores gathered from prompt responses taken before and after DIVAS Seminar I and (2) a two-sample t-test was performed on scripts gathered after the coding workshop and pair programming projects. Subscores within each of the areas of the rubric (Recognize, Analyze, Design, Implement) were evaluated the same way. Effect sizes for significant differences were described by calculating Cohen's d.

## Results and discussion

### Evaluating DIVAS program elements

We measured the impact of DIVAS interventions on participant self-efficacy (SE) in using computation to solve problems, computational thinking (CT), and career path intentions (CI and CK) using instruments described in the previous section. We saw significant gains in self-efficacy over the duration of the program, with SE scores highest following DIVAS Seminar II (Fig 2). Self-efficacy grew steadily throughout the program, increasing by 41.5% from DIVAS Seminar I through the end of pair programming or summer research on average (Cohen's-d = 1.52, Fig 2). Post-workshop, student self-efficacy was maintained throughout the rest of the program. Given that students were asked to solve a number of challenging problems largely independently after the workshop, we see this maintenance of self-efficacy as important.

Average baseline SE scores of DIVAS scholars compared to comparison group 1 (CHM 125 students) and 2 (other workshop participants) are shown in Table 3. Baseline scores for

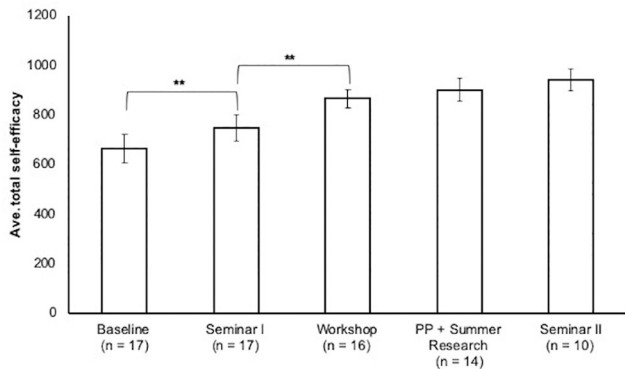

**Fig 2. Average self-efficacy scores (and standard errors) after DIVAS pipeline interventions for the three years of the project.** PP = pair programming. A paired t-test was used to compare means between subsequent interventions. \*\*, p < 0.01.

**Table 3. Baseline SE of DIVAS scholars compared to other groups.**

|  | Average Baseline SE | Std. Error | n |
|---|---|---|---|
| DIVAS scholars | 619 | 51.2 | 14 |
| Comparison group 1 | 484* | 24.5 | 67 |
| Comparison group 2 | 613 (ns) | 52.9 | 15 |
| DIVAS scholars women | 582 | 52.1 | 12 |
| Comparison group 1 women | 440** | 29.4 | 33 |
| Comparison group 2 women | 575 (ns) | 72.8 | 7 |

Significance was determined by a paired t-test. not significant, ns, p < 0.05, *, p < 0.01, **.

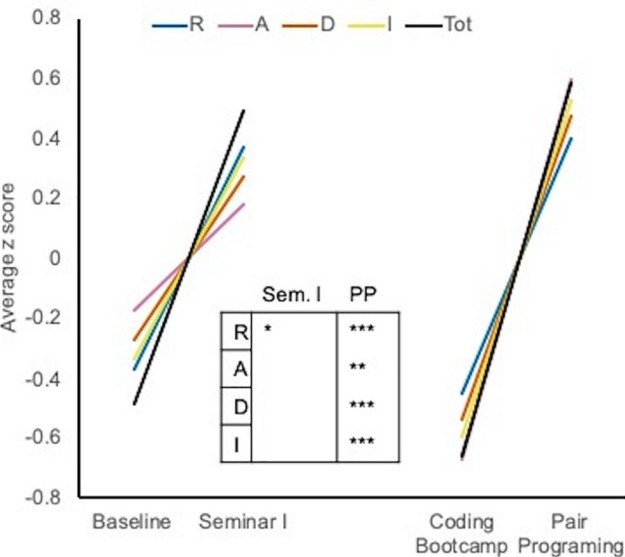

**Fig 3. Average z-scores of total CT score at the indicated DIVAS pipeline interventions.** A paired t-test was used to compare means before and after Seminar I. A two-sample t-test was used to compare means before and after pair programming (PP). *, $p < 0.05$; **, $p < 0.01$; ***, $p < 0001$.

women within each group were also compared. The mean SE of DIVAS scholars (overall and women only) was significantly higher than comparison group 1, but not comparison group 2. The average baseline SE of both the DIVAS scholars and comparison group 2 was comparable to that of first-year CS majors that took the same instrument in a previous study [50]. The average SE of women in our study was lower than the overall group averages (Table 3). This is consistent with other researchers' findings [51–54]. In a 2014 study by Beyer [52], when women were asked to rank their interest in and perceived difficulty of computer science compared to a number of other fields including math, nursing, and English, they ranked computer science as both the most difficult and least interesting field [52]. However, the same study found that women's regard for the field of computer science was higher than that of men and that they had strong agreement with the idea that women have the same ability to succeed in computer science as men [52]. The effects of gender on SE have additionally been shown to become non-significant when accounting for variables such as computer knowledge, experience, and anxiety [51, 52].

As stated in the **Introduction**, we began with the supposition that increased student self-efficacy toward computing was an indicator of growth in computational skill. Average z-scores of total CT scores of participants between baseline and Seminar I and between the coding workshop and pair programming are shown in Fig 3. The 'Computational thinking assessment' subsection of Materials and Methods provides a discussion of the artifacts evaluated using our rubric. Average z-scores in the 'Recognize' category increased after Seminar I (Cohen's-d = 0.82) and all scores increased after pair programming (Cohen's-d = 0.96–1.73, Fig 3). Computational thinking data standardized across the three years of our study is available in S4 Dataset.

Career path interest (CI) and knowledge (CK) was also measured after each intervention in the program (Fig 4). Interest in careers utilizing computing did not change significantly over the course of the program, but CK significantly increased after Seminar I (Cohens-d = 1.31)

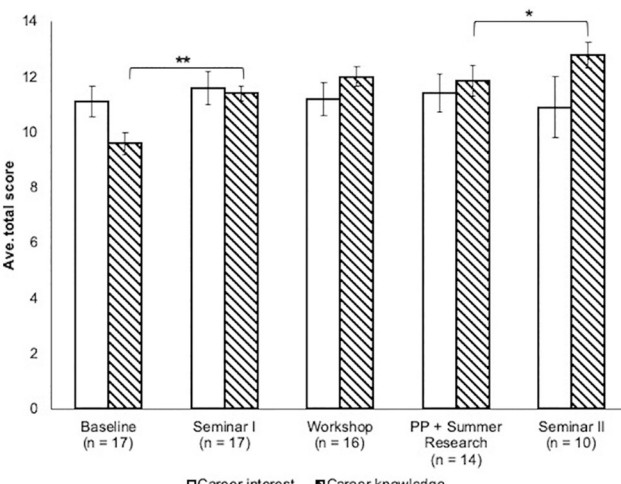

**Fig 4. Average CI and CK scores after DIVAS pipeline interventions for the three years of the project.** Error bars represent standard error of the mean. PP = pair programming. A paired t-test was used to compare means between subsequent interventions for each score type. *, p < 0.05; **, p < 0.01.

and Seminar II (Cohen's-d = 0.52, Fig 4). The standard error in the average CI score was found to increase with each intervention, even as the error in average CK did not.

Overall, every intervention was found to have a positive effect on one or more measures (SE, CK, CT) for at least one of the program years. In the proceeding sections, we discuss the impacts of each intervention separately.

**DIVAS Seminar I.** As described in DIVAS Program Elements, this seminar is the scholar's entry point onto the DIVAS programmatic onramp. We saw significant gains in SE (Cohen's-d = 0.40) and CK (Cohen's-d = 1.31) over the three years of the program (Table 4, Figs 2 and 3).

An additional source of self-efficacy information came from the voluntary completion of an IDEA Student Ratings of Instruction system survey [48], which is conducted at Doane University at the end of each course and that we utilized in DIVAS Seminars I and II. We analyzed self-reported learning gains in the IDEA-defined learning objectives for the eleven scholars who completed the survey. At the end of DIVAS Seminar I, we found that scholars self-reported strong gains in the objectives "Acquiring skills in working with others as a member of a team" and "Learning appropriate methods for collecting, analyzing, and interpreting numerical information." The three cohorts rated both objectives at a median score of 5 out of 5 points. The Doane institutional average over the period of this project on these learning

**Table 4. Average gains in self-efficacy, career interest and career knowledge in DIVAS Seminar I.**

|  | Year 1 | Year 2 | Year 3 | Comb. |
|---|---|---|---|---|
| SE | 57.7 | 36.0 | 167.2 | 82.2** |
| CI | -0.67 | 1.00 | 1.20 | 0.47 |
| CK | 1.67 | 1.67 | 2.20 | 1.82** |
| n | 6 | 6 | 5 | 17 |

Mean change in score from baseline for each year of the program and the three years combined (Comb.) is shown.
**, p < 0.01.

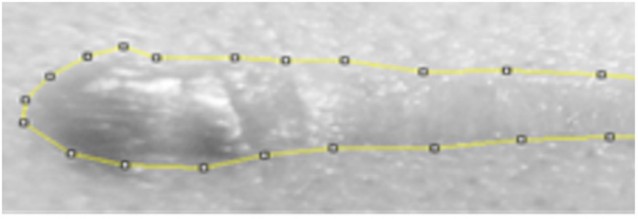

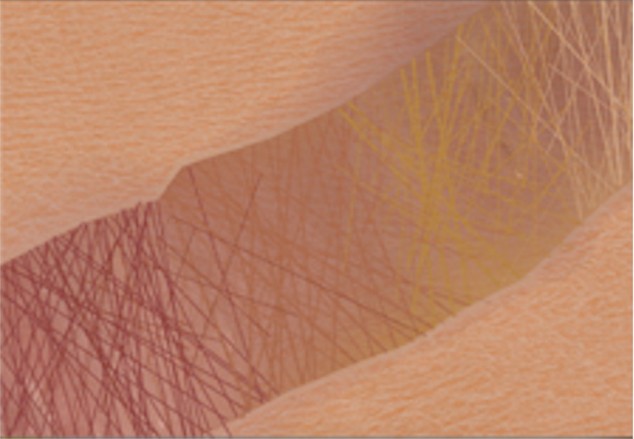

**Fig 5. A collaborative photo journal project.** A DIVAS scholar used ImageJ to analyze images of a healing wound (top) while a design student created a composition depicting the healing process (bottom).

goals were 3.72 and 3.56, respectively. Overall, SE scores improved after scholars completed DIVAS Seminar I, and positively influenced their knowledge of careers utilizing computing skills. Interest in utilizing computing in a future career did not change significantly. Observationally, the seminar was important in building rapport and a shared experience between all members (faculty and students) in the community of practice. In year three, that community was expanded when the DIVAS cohort completed their photo diary project in tandem with 200-level graphic design students (Fig 5). This experience was one of several ways that the seminar served to showcase the multidisciplinary relevance of image collection and analysis.

**Coding workshop.** Modeled after existing Carpentries workshops, the DIVAS workshop was built around participants solving authentic research challenges within a community of practice (see **DIVAS Program Elements**). We gathered self-efficacy and career data pre- and post-workshop. We saw a significant improvement in SE in aggregate over the three years with an average increase of 107.9 points (Cohen's-d = 0.57, $p < 0.01$, Table 5). There were no significant changes in CI or CK over the three-year period.

**Table 5. Average gains in self-efficacy, career interest, and career knowledge in the coding workshop.**

|      | Year 1 | Year 2 | Year 3 | Comb. |
|------|--------|--------|--------|-------|
| SE   | 91.4   | 131.7  | 96.0   | 107.9[***] |
| CI   | -0.20  | -0.50  | -1.60  | -0.75 |
| CK   | 0.60   | 0.50   | 0.60   | 0.56  |
| n    | 5      | 6      | 5      | 16    |

Mean change in score from Seminar I for each year of the program and the three years combined (Comb.) is shown.
[*], $p < 0.5$.

**Table 6. Participant responses to the question 'What percentage of the day's material do you feel you have mastered?' for each day of the coding workshop.**

|  | Day 1 | Day 2 | Day 3 |
|---|---|---|---|
| Python Intro | 75.9% | 76.3% | - - - |
| Image Processing | 63.3% | 63.0% | 72.2% |

n = 31–40.

At the end of each day of the workshop, we also asked participants to rate the percentage of the day's material they felt they had mastered. Data was compiled for all participants, including those who were not DIVAS scholars. We found a high average perceived mastery for the Python/Bash/git portion of the workshop, and then a drop for the first two days of the image processing portion (Table 6). We believe this is due to the increased complexity in the subject matter. By the third day of the image processing portion, this metric rose as participants were able to use their newfound skills to complete the challenge questions successfully.

We found the coding workshop format to be effective as it immersed scholars in an enriching skill development environment. Though coding training was intensive, the participants' self-reported improvements in mastery support the observation that scholars see tangible benefits from their persistence. The workshop also provided two cycles of challenge, learn, and achieve—in the spirit of Challenge Based Learning [55]—to provide participants multiple opportunities to struggle with new concepts and see rewards.

**Pair-programming projects and independent research.** Following the coding workshop, scholars employed pair programming to solve a colorimetric and a morphometric image analysis problem (Fig 6, Table 1, S8 File) as described in DIVAS Program Elements.

Following pair programming work, scholars had the opportunity to complete three or more weeks of independent research (required in year 1, optional in years 2 and 3). Scholars either chose to work on existing projects, or design their own, within a faculty mentored research group. Scholars were supported in this research through existing grants and institutional summer research programs. Examples of scholars' projects included locating breaks in veterinary x-ray images, processing and solving printed Sudoku problems, greatly improving program performance by translating Python scripts into parallelized C++ code, and measuring chemotaxis of bacteria toward potential attractants (Table 1).

We collected SE and career data upon the conclusion of summer research, if the scholar participated, or at the conclusion of pair programming projects for those students not

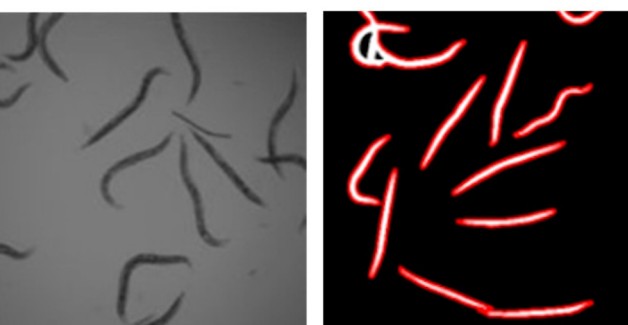

**Fig 6. Example pair programming project in which students aimed to count the dead (straight) worms in a series of images.** The unprocessed image is from [56].

participating in research. Similar to the coding workshop, we did not observe changes in CI or CK. Although self-efficacy moved in a positive direction, we did not see significant gains. This was not completely unexpected, however, since students' self-efficacy was already high following the coding workshop (Fig 2). Again, we see the maintenance of SE throughout this programmatic period as significant given the increased complexity of problems and independent work students were given.

Observationally, the pair programming and summer research projects were where scholars truly experienced the team-based environment of computational work. They learned to leverage each other's ideas and expertise to develop approaches to solving a variety of problems. We found that scholars tended to work amongst themselves *before* seeking input from one of the faculty mentors. We considered this a healthy development of independence and teamwork that reflected the confidence scholars gained in their individual and collective skill sets. We also observed cases where one or more scholars would be given special authority by the group. While this was often productive, we also observed that it sometimes contributed to an over/under-functioning dynamic between pairs. Because of this, we were especially mindful of giving praise for taking risks and highlighting the specific strengths of each project and each scholar separately. We also worked to minimize this over/under-functioning dynamic when selecting pairs for each project so as to maximize each student's engagement.

**DIVAS Seminar II.** DIVAS Seminar II marks the end to the program and occurred each spring, concurrent with Seminar I, as described in the DIVAS Program Elements. The last year of the seminar occurred during the first wave of the COVID-19 pandemic, resulting in a response rate to the survey that was too low to report. For years 1 and 2, no significant gain in SE or CI from the summer was observed, but similar to Seminar I, we did see a gain in CK (Cohen's-d = 0.52, Table 7).

Similar to DIVAS Seminar I, responses on the IDEA survey for DIVAS Seminar II were also analyzed for perceived learning gains. Survey data showed that students perceived the largest gains in "Learning appropriate methods for collecting, analyzing, and interpreting numerical information" (4.33 ± 0.52). Scholars also responded positively to the statement, "My background prepared me well for this course's requirements" (4.5 ± 0.55). All scholars rated these three items at either a '4' or '5' on a 5-point scale.

## Project outcomes and next steps

Over the three years of the project, scholars experienced significant increases in SE towards computing from the beginning of Seminar I to the end of summer programming (Fig 2). The most significant gains (p < 0.05) occurred during Seminar I (Cohen's-d = 0.40) and the coding workshops (Cohen's-d = 0.67). The impact of the DIVAS program on scholars' career interest and knowledge was more subtle. Although scholars did not show significant gains in CI gain,

**Table 7. Average gains in self-efficacy, career interest, and career knowledge in DIVAS Seminar II.**

|     | Year 1 | Year 2 | Comb. |
| --- | --- | --- | --- |
| SE | -11.3 | 56.0 | 22.4 |
| CI | -0.25 | 0.75 | 0.25 |
| CK | 2.25 | 0.25 | 1.25* |
| n | 4 | 4 | 8 |

Mean change in score from PP/summer research for each year of the program and the two years combined (Comb.) is shown.

*, p < 0.5.

gains in CK were seen in both Seminars I and II (Fig 4, Tables 4 and 7), both of which include explicit career exploration components. Scholars were also observed to become 'warmer' or 'colder' to a career utilizing computing as they moved through the program. This effect is apparent in the increased standard error in post-intervention CI scores, which started at ± 0.55 after Seminar I, grew to ± 1.1 after Seminar II. This same trend was observed in analysis of only the subset of scholars that we had full data (S1 Dataset) for from Seminar I through Seminar II. Overall, we see this as an encouraging progression, especially because scholar self-efficacy grew steadily throughout the program.

Given the small samples in our study, it is encouraging that our results are consistent with those of previous studies that have used the same instruments. In a study of students enrolled in introductory computer science (CS) courses, all groups except CS majors in a special honors program showed declines in their interest in pursuing careers involving computation, similar to our findings [39]. An important positive benefit in interest in computational careers is improved retention in CS courses [40]. However, a weaker, but additional highly significant predictor of retention within CS courses is acquiring information about careers involving computational skills (Peteranetz 2018). We saw an increase in CI for some scholars, but an overall increase in CK over the course of our program (Fig 4, Tables 4 and 7). Previous research has shown that being able to immediately apply what is being learned positively impacts student self-efficacy in computing [38]. Increased self-efficacy further promotes skill development [57]. The steady growth in self-efficacy observed in our program is consistent with these previous findings given the frequent immediate, developmentally appropriate opportunities for application of knowledge that are built into it. Also consistent with previous SE and skills linkage observations, are our observations that improvement in computational skill occurs with improvement in self-efficacy (Figs 2 and 3).

Retention in STEM is often used to indicate the success of engagement programs such as DIVAS. A large majority (82%) of scholars entered the program as first year students, taking Seminar I in the second semester of their first year and Seminar II the second semester of their sophomore year. All DIVAS scholars were still majoring in STEM fields one year after taking DIVAS II Seminar. All but one scholar was retained throughout the entire one-year program for a retention rate of 94%. This is in contrast to first-to-second year STEM retention rates of approximately 72% and 56% at DU and SEU, respectively. In a number of ways, DIVAS scholars have also persisted in coding and have incorporated their new skills into their academic careers, extracurricular activities, and career planning. One scholar majoring in biology declared a minor in software development. A second biology major switched to a bioinformatics major, and two scholars have taken non-required electives that emphasize computational skills. One scholar participated in an external REU program in computational and systems biology, and eight have elected to conduct research projects that incorporate coding or computational thinking. Three DIVAS scholars have worked as peer tutors for DU's CCLA. One former scholar is pursuing a Ph.D. in chemical biology with a significant computational component to their research and another student who participated in both the coding workshop and paired programming is pursuing a Ph.D. in complex biosystems.

Overall, even given the small sample represented in this study, we see great potential in the DIVAS approach of introducing novice students to computing through visual media within a community of practice. A significantly higher percentage of DIVAS scholars identified as women (76%) than the total percentage of women in the STEM majors at either DU or SEU, as well as women in the full-time science and engineering (S&E) or S&E-related occupations combined (51.7%) [58]. However, the small sample in this study leaves open the possibility that confounding factors may have led to the trends we observed. For example, the baseline

self-efficacy of DIVAS scholars is significantly higher than students surveyed from an introductory chemistry, including among women only (Table 3). It is possible that in our study we selected for students who were already predisposed toward the outcomes we observed above. Additional study through the implementation of DIVAS program elements in a broader array of educational contexts on a larger scale will make it possible to confirm our pilot study findings or reveal new trends. To this end, we hope by presenting our preliminary results and providing programmatic support materials, partnerships will be formed that will enable an expanded study on the efficacy of the DIVAS approach.

## Supporting information

**S1 File. DIVAS Seminar I syllabus example.**
(PDF)

**S2 File. DIVAS Seminar I photo diary assignment.**
(PDF)

**S3 File. DIVAS Seminar I weekly discussion prompts examples.**
(PDF)

**S4 File. DIVAS Seminar II syllabus example.**
(PDF)

**S5 File. DIVAS Seminar II Bash/git review and repo curation.**
(PDF)

**S6 File. DIVAS Seminar II introduction to parallelism presentation.**
(PDF)

**S7 File. DIVAS Seminar II burning ship assignment.**
(PDF)

**S8 File. Pair programming project example.**
(PDF)

**S9 File. DIVAS career path and self efficacy instrument.**
(PDF)

**S10 File. Computational thinking rubric draft 2.** This rubric was used in year 1 of the study.
(PDF)

**S11 File. Computational thinking rubric draft 3.** This rubric was used in year 2 of the study.
(PDF)

**S12 File. Computational thinking rubric draft 4.** This rubric was used in year 3 of the study.
(PDF)

**S13 File. Computational thinking rubric changes from draft 2 to draft 3.**
(PDF)

**S14 File. Computational thinking rubric changes from draft 3 to draft 4.**
(PDF)

**S15 File. Cup stacking prompt configuration 1.**
(PDF)

**S16 File. Cup stacking prompt configuration 2.**
(PDF)

**S17 File. Cup stacking prompt configuration 3.**
(PDF)

**S1 Dataset. Career path and self-efficacy data.**
(XLSX)

**S2 Dataset. IDEA survey data.**
(XLSX)

**S3 Dataset. Coding workshop survey data.**
(XLSX)

**S4 Dataset. CT rubric scorings.**
(XLSX)

**S5 Dataset. Baseline SE comparisons.**
(XLSX)

## Acknowledgments

We would like to thank Adam Erck for his leadership in instituting the CCLA, Dr. Chris Huber for allowing us to use his introductory chemistry course for this study, and Sarah Zulkoski for her tireless advocacy and support of this project.

## Author Contributions

**Conceptualization:** Tessa Durham Brooks, Raychelle Burks, Erin Doyle, Mark Meysenburg, Tim Frey.

**Data curation:** Tessa Durham Brooks, Erin Doyle, Mark Meysenburg, Tim Frey.

**Formal analysis:** Tessa Durham Brooks, Erin Doyle, Mark Meysenburg, Tim Frey.

**Funding acquisition:** Tessa Durham Brooks, Raychelle Burks, Erin Doyle, Mark Meysenburg.

**Investigation:** Tessa Durham Brooks, Raychelle Burks, Erin Doyle, Mark Meysenburg, Tim Frey.

**Methodology:** Tessa Durham Brooks, Raychelle Burks, Erin Doyle, Mark Meysenburg, Tim Frey.

**Project administration:** Tessa Durham Brooks, Raychelle Burks, Erin Doyle, Mark Meysenburg.

**Resources:** Raychelle Burks, Erin Doyle, Mark Meysenburg.

**Software:** Erin Doyle, Mark Meysenburg.

**Supervision:** Tessa Durham Brooks, Raychelle Burks, Mark Meysenburg.

**Visualization:** Tessa Durham Brooks.

**Writing – original draft:** Tessa Durham Brooks, Raychelle Burks.

**Writing – review & editing:** Tessa Durham Brooks, Raychelle Burks, Erin Doyle, Mark Meysenburg, Tim Frey.

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
