## [Decision Letter · Decision Letter 0]

2 Dec 2020

PONE-D-20-33866

Digital Imaging and Vision Analysis in Science Project improves the self-efficacy and skill of undergraduate students in computational work

PLOS ONE

Dear Dr. Durham Brooks and co-authors:

Thank you for submitting your manuscript to PLOS ONE. After careful consideration, we feel that it has merit but does not fully meet PLOS ONE’s publication criteria as it currently stands. Therefore, we invite you to submit a revised version of the manuscript that addresses the points raised during the review process.

As revising the manuscript, authors need to specifically improve the full points by all three reviewers as below. As reviewer3 pointed out, authors need to pay attention to small sample size, and possibly explain this as a limitation in the Discussion section.

We look forward to receiving your revised manuscript.

Kind regards,

G Hacisalihoglu

Academic Editor

PLOS ONE

Journal Requirements:

Reviewers' comments:

Reviewer's Responses to Questions

**Comments to the Author**

1. Is the manuscript technically sound, and do the data support the conclusions?

Reviewer #1: Partly

Reviewer #2: Partly

Reviewer #3: Partly

2. Has the statistical analysis been performed appropriately and rigorously? 

Reviewer #1: No

Reviewer #2: Yes

Reviewer #3: I Don't Know

3. Have the authors made all data underlying the findings in their manuscript fully available?

Reviewer #1: No

Reviewer #2: Yes

Reviewer #3: Yes

4. Is the manuscript presented in an intelligible fashion and written in standard English?

Reviewer #1: Yes

Reviewer #2: Yes

Reviewer #3: Yes

5. Review Comments to the Author

Reviewer #1: The DIVAS program is interesting and seems like a quality program worth sharing with the field. The article as a whole is well written and enjoyable to read. My detailed comments are below, grouped as "major," "minor," and "typos/tiny."

Major issues/comments

The soundness of including t-tests for individual years are questionable with how small the sample sizes are. Unless there are substantive reasons to separate the years (as it seems there were in the DIVAS II seminar), it should all be analyzed together. Also, full t-test stats should be reported, not just effect sizes.

Figure 4 is incredibly useful and is underused. It would be helpful to introduce it with Table 3 and reference it every time self-efficacy results are presented. I think it would also be good to have parallel figures for the career path and rubric data.

Based on the heading formats, it seems like Computational Thinking is a sub-heading under Self-efficacy and Career Path Data by Intervention, but logically that doesn’t make sense. Relatedly, the length given to the self-efficacy and career path results seems quite disproportionate compared to the computational thinking results and overall outcomes. Part of the length discrepancy seems to be due to the fact that the Self-efficacy and Career Path Data by Intervention section contains a lot of information that would fit better in the earlier sections describing the program (lines 120-218).

Minor issues/comments/questions

Was recruitment specifically targeted at women and URM students? If so, how?

About reliability of rubric data: What version of IRR was used? Kappa, % agreement, etc. (line 281). Were the raters the same for every cohort and every part of the program? Were all artifacts scored by multiple raters or just the ones used to estimate reliability? (After reviewing the supplemental file, I see the answer, but it would be helpful to have a statement about it in the text.) If all were scored by more than one rater, were alpha estimates calculated with all data or only a subset (so that each artifact had only one set of scores in the data)?

Also about the rubric data, what is the significance of having one record for 2 scholars for some pair programming entries but individual records for other pair programming entries? And how was this handled in the analysis? Pair scores are not directly comparable to individual scores at different time points.

The career path measure in a combination of two other scales (information on careers and career aspirations). What’s the justification for combining them into a single scale?

Please clarify how the workshop worked with regard to attendees not in the program. Was it a workshop open to all that was part of the program? Or were there individuals who started but did not complete anything after the workshop? Or something else?

Were the summer activities connected to the institution, e.g., done for academic credit, etc.?

Given the large number of analyses, I recommend you use a correction for Type I error.

The IDEA data included in the results should either be first described in the method section or not included. Also, these data are not provided in the supplemental files.

Line 352 – the comments about the photo diary project seem out of place. I think it would fit better earlier when the different program components are being described.

There is some redundancy throughout the paper, especially with describing the sample (e.g., the fact that the sample was 76% women is included 5 times if you count the abstract). Most of the first paragraph under “Results and Discussion” is redundant with what is presented in the “Study Context” section.

I would like to see the Results and Discussion explicitly tie back to the research questions

Typos/tiny issues

As written, research question 3 (lines 209-210) does not make sense.

The abbreviation “fig” is written/punctuated at least 3 different ways.

Capitalization within headings is inconsistent.

Please define the acronym REU the first time it is used.

CT_rubrics_scoring xlsx file: $CT Assessments Table(B2:C2) appears to have a draft comment in it.

What is the significance of NA vs. blanks vs question marks in the CT_rubrics_scoring file? If they are all indications of missing data (intentional or not), consider using the same method of coding throughout.

Reviewer #2: The manuscript is technically sound but does not have enough data to form conclusions. This is the beginning of a good pilot study. Statistical analyses are sound. I believe relavent data were provided though very difficult to obtain on the stated website. The manuscript is well-written in standard English. Several issues are noted in the Comments provided in the attached PDF.

Reviewer #3: Overall this manuscript describes an interesting project aimed at improving undergraduates’ computational skills and increasing their awareness of and interest in the use of computation in the life sciences. While the project is interesting and potentially valuable the research described in the manuscript is lacking in a number of key areas.

The authors focus on two hypotheses:

1) The DIVAS Project will impact student self-efficacy, computational competency, and career path interest.

2) That as participants become more familiar with computational tools, they will additionally show more interest in career paths that would utilize said tools.

The difficulty in evaluating the results contained in this manuscript is related to several factors which I have grouped together into two categories.

Category 1 Sample Size etc.

The results are based on an extremely small sample size, self-selection of the students (they had to have some interest to apply), no information about the demographics of the potential applicant pool or how this relates to the actual participants. The number of women and URM students is impressive, but without knowing the demographics of the potential applicant pool it is impossible to evaluate the meaning of this.

Category 2 Career Path.

The authors combined all data for all the survey questions related to career path into a single score. The difficulty with this is that of the 7 questions related to career path on the survey only 3 of them focus on interest and the other 4 relate to knowledge about career paths. The result is that the combined data do not directly address their hypotheses. It is unclear if just focusing on the 3 career interest questions would generate the similar results to those reported for the combined data. Additionally, the authors state that “Scholars were also observed to become ‘warmer’ or ‘colder’ to a career utilizing computing as they moved through the program. This effect is apparent in the increased standard deviation in post-intervention career interest scores, which started at ± 2.3 after Seminar I, grew to ± 3.02 after the coding workshop, and increased to ± 3.46 after pair programming/summer research. We see this as an encouraging progression, especially because scholar self-efficacy grew steadily throughout the program.” The increased standard deviation suggests that reporting how many students increased their interest vs how many decreased their interest would be very valuable.

Category 3 Control Comparisons

The authors make no effort to compare their results to other similar programs or to include any form of a control group. Comparisons such as; are any of the improvements that they saw in student self-efficacy greater than, less than or equal to that seen in other programming workshops, or how many of the students not in this program become engaged in research projects that involve computational approaches, would be very useful.

My suggestion to the authors is to view the data they have as a qualitative study and describe the outcomes that they observed rather than trying to make this a quantitative study based on a very small sample. Descriptions of the impact of this experience on the participants may be more impactful than the quantitative descriptions with no comparisons. As I said initially the program is interesting and may be having a positive impact, but based on the data presented it is very difficult to know the degree of impact that this program is having on the students.

6. PLOS authors have the option to publish the peer review history of their article (what does this mean?). If published, this will include your full peer review and any attached files.

Reviewer #1: No

Reviewer #2: No

Reviewer #3: No

---

## [Author Response · Author response to Decision Letter 0]

2 Apr 2021

1. Please ensure that your manuscript meets PLOS ONE's style requirements, including those for file naming. -- Corrected

2. Please provide additional details regarding participant consent. -- Corrected in revised manuscript

3. Please include captions for your Supporting Information files at the end of your manuscript, and update any in-text citations to match accordingly. -- Corrected

The soundness of including t-tests for individual years are questionable with how small the sample sizes are. Unless there are substantive reasons to separate the years (as it seems there were in the DIVAS II seminar), it should all be analyzed together. Also, full t-test stats should be reported, not just effect sizes. -- Addressed statistical approach in the response letter. p-values and effect sizes were provided for every significant difference

Figure 4 is incredibly useful and is underused. It would be helpful to introduce it with Table 3 and reference it every time self-efficacy results are presented. I think it would also be good to have parallel figures for the career path and rubric data. -- New figures were created in revised manuscript

Based on the heading formats, it seems like Computational Thinking is a sub-heading under Self-efficacy and Career Path Data by Intervention, but logically that doesn’t make sense. Relatedly, the length given to the self-efficacy and career path results seems quite disproportionate compared to the computational thinking results and overall outcomes. Part of the length discrepancy seems to be due to the fact that the Self-efficacy and Career Path Data by Intervention section contains a lot of information that would fit better in the earlier sections describing the program (lines 120-218). -- a major reorganization and consolidation of information is in the revised manuscript

Was recruitment specifically targeted at women and URM students? If so, how? -- Addressed in revised manuscript

About reliability of rubric data: What version of IRR was used? Kappa, % agreement, etc. (line 281). Were the raters the same for every cohort and every part of the program? Were all artifacts scored by multiple raters or just the ones used to estimate reliability? (After reviewing the supplemental file, I see the answer, but it would be helpful to have a statement about it in the text.) If all were scored by more than one rater, were alpha estimates calculated with all data or only a subset (so that each artifact had only one set of scores in the data)? -- Details regarding rubric development and scoring was added to the revised manuscript

Also about the rubric data, what is the significance of having one record for 2 scholars for some pair programming entries but individual records for other pair programming entries? And how was this handled in the analysis? Pair scores are not directly comparable to individual scores at different time points. -- A description of this analysis was provided in the main manuscript. There are two pair programming files for each scholar. If the scholar was paired with a non-scholar, as occurred in years 1 and 2, the non-scholar did not appear more than once.

The career path measure in a combination of two other scales (information on careers and career aspirations). What’s the justification for combining them into a single scale? -- This was justified by our reliability analysis. However, we separated out these scales based on how they were used previous.

Please clarify how the workshop worked with regard to attendees not in the program. Was it a workshop open to all that was part of the program? Or were there individuals who started but did not complete anything after the workshop? Or something else? -- This was clarified in the revised manuscript

Were the summer activities connected to the institution, e.g., done for academic credit, etc.? -- This was clarified in the revised manuscript

Given the large number of analyses, I recommend you use a correction for Type I error. -- This was addressed in the response letter

The IDEA data included in the results should either be first described in the method section or not included. Also, these data are not provided in the supplemental files. -- Corrected

Line 352 – the comments about the photo diary project seem out of place. I think it would fit better earlier when the different program components are being described. -- This was addressed in the revised manuscript

There is some redundancy throughout the paper, especially with describing the sample (e.g., the fact that the sample was 76% women is included 5 times if you count the abstract). Most of the first paragraph under “Results and Discussion” is redundant with what is presented in the “Study Context” section. -- the manuscript was significantly restructured to address this comment

I would like to see the Results and Discussion explicitly tie back to the research questions -- This has been addressed in the revised manuscript

As written, research question 3 (lines 209-210) does not make sense. -- Revised

The abbreviation “fig” is written/punctuated at least 3 different ways. -- Corrected

Capitalization within headings is inconsistent. -- Corrected

Please define the acronym REU the first time it is used. -- Corrected

CT_rubrics_scoring xlsx file: $CT Assessments Table(B2:C2) appears to have a draft comment in it. -- Corrected

What is the significance of NA vs. blanks vs question marks in the CT_rubrics_scoring file? If they are all indications of missing data (intentional or not), consider using the same method of coding throughout. -- Clarified in the supplementary material

Several issues are noted in the Comments provided in the attached PDF. -- These comments were incorporated into the revised manuscript, in particular making the preliminary nature of the data more explicit

Category 1 Sample Size etc.

The results are based on an extremely small sample size, self-selection of the students (they had to have some interest to apply), no information about the demographics of the potential applicant pool or how this relates to the actual participants. The number of women and URM students is impressive, but without knowing the demographics of the potential applicant pool it is impossible to evaluate the meaning of this. -- We added a section analyzing baseline self-efficacy between scholars and a population representing the applicant pool. We added more information about the demographics of the applicant pool

Category 2 Career Path.

The authors combined all data for all the survey questions related to career path into a single score. The difficulty with this is that of the 7 questions related to career path on the survey only 3 of them focus on interest and the other 4 relate to knowledge about career paths. The result is that the combined data do not directly address their hypotheses. It is unclear if just focusing on the 3 career interest questions would generate the similar results to those reported for the combined data. -- This was address by splitting out the two sets of questions

Additionally, the authors state that “Scholars were also observed to become ‘warmer’ or ‘colder’ to a career utilizing computing as they moved through the program. This effect is apparent in the increased standard deviation in post-intervention career interest scores, which started at ± 2.3 after Seminar I, grew to ± 3.02 after the coding workshop, and increased to ± 3.46 after pair programming/summer research. We see this as an encouraging progression, especially because scholar self-efficacy grew steadily throughout the program.” The increased standard deviation suggests that reporting how many students increased their interest vs how many decreased their interest would be very valuable. -- This information was added to the revised manuscript

Category 3 Control Comparisons

The authors make no effort to compare their results to other similar programs or to include any form of a control group. Comparisons such as; are any of the improvements that they saw in student self-efficacy greater than, less than or equal to that seen in other programming workshops, or how many of the students not in this program become engaged in research projects that involve computational approaches, would be very useful. -- This was addressed in the revised manuscript

My suggestion to the authors is to view the data they have as a qualitative study and describe the outcomes that they observed rather than trying to make this a quantitative study based on a very small sample. Descriptions of the impact of this experience on the participants may be more impactful than the quantitative descriptions with no comparisons. As I said initially the program is interesting and may be having a positive impact, but based on the data presented it is very difficult to know the degree of impact that this program is having on the students. -- This was addressed in the revised manuscript and response letter

---

## [Editor Report · Decision Letter 1]

12 Apr 2021

Digital Imaging and Vision Analysis in Science Project improves the self-efficacy and skill of undergraduate students in computational work

PONE-D-20-33866R1

Dear Dr. Brooks,

We’re pleased to inform you that your manuscript has been judged scientifically suitable for publication and will be formally accepted for publication once it meets all outstanding technical requirements.

Kind regards,

Gokhan Hacisalihoglu

Academic Editor

PLOS ONE
---

## [Editor Report · Acceptance letter]

27 Apr 2021

PONE-D-20-33866R1 

Digital Imaging and Vision Analysis in Science Project improves the self-efficacy and skill of undergraduate students in computational work 

Dear Dr. Durham Brooks:

I'm pleased to inform you that your manuscript has been deemed suitable for publication in PLOS ONE. Congratulations! Your manuscript is now with our production department. 

Kind regards, 

on behalf of

Professor Gokhan Hacisalihoglu 

Academic Editor

PLOS ONE